# Health and Safety Reps in COVID-19—Representation Unleashed?

**DOI:** 10.3390/ijerph20085551

**Published:** 2023-04-18

**Authors:** Sian Moore, Minjie Cai, Chris Ball, Matt Flynn

**Affiliations:** 1Greenwich Business School, University of Greenwich (Maritime Campus), London SE10 9LS, UK; 2School of Business, University of Leicester, Leicester LE2 1RQ, UK

**Keywords:** COVID-19, occupational health and safety, trade unions, worker representation, joint committees

## Abstract

The paper explores the role of UK union health and safety representatives and changes to representative structures governing workplace and organisational Occupational Health and Safety (OHS) during COVID-19. It draws upon a survey of 648 UK Trade Union Congress (TUC) Health and Safety (H&S) representatives, as well as case studies of 12 organisations in eight key sectors. The survey indicates expanded union H&S representation, but only half of the respondents reported H&S committees in their organisations. Where formal representative mechanisms existed, they provided the basis for more informal day-to-day engagement between management and the union. However, the present study suggests that the legacy of deregulation and the absence of organisational infrastructures meant that the autonomous collective representation of workers’ interests over OHS, independent of structures, was crucial to risk prevention. While joint regulation and engagement over OHS was possible in some workplaces, OHS in the pandemic has been contested. Contestation challenges pre-COVID-19 scholarship suggestingthat H&S representatives had been captured by management in the context of unitarist practice. The tension between union power and the wider legal infrastructure remains salient.

## 1. Introduction

In the UK, workplace health and safety (H&S) representatives have legal rights to represent the interests and concerns of workers over health and safety, to make representations on potential hazards and dangers and to have contact with health and safety inspectors from the Health and Safety Executive or the local authority [1]. Trade unions have the right to appoint H&S representatives where they are recognised. In a pandemic, their independence from and relationship with management is potentially critical in assessing and addressing risk for workers and for public health. Yet, pre-COVID-19 scholarship suggested that the regulatory model of worker representation on health and safety may have lost relevance in the light of the reduced coverage of trade unions [2,3,4]. Moretta et al. suggest that UK Occupational Health and Safety (OHS) is largely defined by self-regulation and that ‘the COVID-19 crisis has shown the UK’s workplace health and safety enforcement regime to be ‘a hollowed-out shell’ [5] (p. 16). In previous work, we pointed to the significance of political economy for worker health and safety during COVID-19, the impact of de-regulation of OHS, the contraction in state support for collective bargaining, the influence of government migration policy, and the implications of diminishing access to both occupational and statutory sick pay [6]. There is an integral relationship between workplace health and safety, independent worker representation, and public health.

The present study explores the role of UK union H&S representatives and the operation of representative structures and mechanisms governing workplace and organisational OHS during COVID-19. It does so in a context in which the capacity of representatives has been undermined by the legacy of deregulation [7], and where non-standard workers are excluded from representation over OHS and rights to sick pay. As in previous studies of OHS in the pandemic [6], the research points to the absence of H&S committees, even in workplaces and organisations with union H&S representation. This discrepancy prompts discussion of Walters and Wadsworths’ [8] notion of the autonomous collective representation of workers’ interests in health and safety. They propose that despite formal tripartite arrangements, and even where there are formal representative infrastructures at the organisational level, OHS has become unitarist in practice and there has been a move towards the capture and incorporation of H&S representatives by management, although exceptionally there may be autonomous collective activity [8]. Drawing upon a survey of UK Trade Union Congress (TUC) H&S representatives and case-studies of 12 organisations in eight key sectors, we tested practice in the extreme circumstances of the pandemic and where there was some form of collective representation over OHS. In terms of pluralism, the survey results suggest some resurgence of joint regulation. Yet, both survey and case studies demonstrate the proactive role played by union H&S representatives and informal day-to-day engagement with management, but, also, the restoration of OHS as an arena for conflict. In illuminating the agency of H&S representatives that is independent of structures, the research confirms Hall and Tuckers’ evocation of the tension between union power and the OHS regulatory framework [9]. 

## 2. Context

Pre-COVID-19 studies have shown that worker health and safety, including psycho-social safety climate, is related to population health and to organisational performance and productivity at the national level [10]. With regard to the workplace, research has demonstrated the importance of effective and autonomous trade union representation as ‘an essential monitoring and correcting mechanism to effectively reduce risks at work’ [11]. The influence of union safety representatives (or similar) has been proved to achieve more effective OHS, while awareness of health and safety representation makes a difference to self-reported preventive action by workers [12]. Sojourner and Yang [13] found that in the US, union victory in close National Labor Relations Board (NLRB) certification elections is associated with increased occupational safety law enforcement activity as well as increased worker representation in the enforcement process in terms of initiating complaints to and subsequent inspections by the Occupational Safety and Health Administration. However, they suggest that declining unionisation rates in the US are reducing workers’ capacity to exercise their rights and to participate in occupational safety enforcement. 

Prior to the pandemic, Walters and Wadsworth [8] proposed that across Europe, institutional forms of H&S representation are largely conceived in pluralist industrial relations contexts, but operationalized in more unitary terms with direct consequences for the structure and operation of organisational H&S relationships. Their study indicated an increase in more direct forms of worker involvement in OHS, with representatives incorporated into safety arrangements that are controlled by managers, and with representatives communicating and monitoring managers’ messages. 

Research on COVID-19 can underplay the role of the workplace as a key site of infection and prevention, and large statistical surveys may omit economic risk as a key factor predicting behaviour. There is limited consideration of the role of the Health and Safety Executive, trade unions, and workplace representatives in risk assessment and the protection of worker health and safety, especially their impact on infection beyond the workplace. However, initial research on the role of unions in COVID-19 has indicated that they may act as key agents in public health as well as defending labour market conditions. Han finds that in the US, union workers experienced greater job security than non-union workers in the pandemic, and where they were laid off, they were more likely to receive pay. Unions also reduced workplace inequality for women and workers of colour prior to and during the pandemic [14]. Focusing on health and safety, Taylor and Chan described ‘the last instance of successful independent union mobilisation’ in Hong Kong (before the regime cracked down), demanding that the health authority and government provide protective equipment to hospital workers in the early stages of the pandemic [15] (p. 711). Hall and Tucker documented union attempts to influence government policy and workplace COVID-19 plans in Ontario, but suggest significant constraints in the absence of institutionalized channels for union voice above the workplace level [9]. In the UK, action by workers in the Driver and Vehicle Licensing Authority (DVLA) and by the National Education Union (NEU) to protect their members are notable. 

This study uses a mixed methods approach to ask how far COVID-19 led to changes in organisational and sectoral mechanisms, processes for worker representation and joint regulation on health and safety. Drawing on Walters and Wadsworth [8], the study distinguishes between formal OHS representative infrastructures informed by pluralism, and organisational and workplace practice in the context of the pandemic, while acknowledging the dynamic between structure and practice. In identifying unitarist practice prior to COVID-19, Walters and Wadsworth suggested three trends: firstly, limitations on management support for representation; secondly, the marginalisation of unions and move towards direct representation; and, thirdly, the incorporation of H&S representatives into systems controlled by management. After discussing methods, the paper looks at evidence of changes to structures of representation during the pandemic and then to the operation of such relationships at sectoral, organizational, and workplace levels, focusing on the agency of representatives. Discussion and conclusions reflect on the notion of autonomous collective OHS representation and tension between union organisation and the OHS regulatory framework.

Drawing on the literature and key expert interviews, five hypotheses were developed to test whether the presence of a health and safety committee with trade union representation was associated with collective organisation, activity, and perceived risk. Secondly four complementary hypotheses were developed to test the impact of increased H&S representation during the pandemic, a variable that may indicate more autonomous activity *independent of H&S committees*. Pearson correlations were used to test associations between variables.

**Hypotheses 1 (H1).** *The presence of a health and safety committee with trade union representation is positively associated with increased H&S representation*. 

**Hypotheses 2 (H2a).** *The presence of a health and safety committee with trade union representation is positively associated with negotiation or consultation over OHS issues during the pandemic*. 

**Hypotheses 2 (H2b).** *Increased H&S representation is negatively associated with negotiation or consultation over OHS issues during the pandemic*.

**Hypotheses 3 (H3a).** *The presence of a health and safety committee with trade union representation is positively associated with the frequency of engagement between trade union representatives and managers during the pandemic*.

**Hypotheses 3 (H3b).** *Increased H&S representation is positively associated with the frequency of engagement between trade union representatives and managers during the pandemic*.

**Hypotheses 4 (H4a).** *The presence of a health and safety committee with trade union representation is negatively associated with perceived OHS risk*. 

**Hypotheses 4 (H4b).** *Increased H&S representation is negatively associated with perceived OHS risk*.

**Hypotheses 5 (H5a).** *The presence of a health and safety committee with trade union representation is positively associated with perceived managerial support for H&S representation*.

**Hypotheses 5 (H5b).** *Increased H&S representation is negatively associated with perceived managerial support for H&S representation*. 

## 3. Materials and Methods

The research was funded under the UKRI ESRC scheme, Ideas to address COVID-19. A mixed methods approach was taken, integrating qualitative and quantitative data collection and analysis in parallel phases. Interpretations are based on the combined strengths of three sets of data, providing a more nuanced understanding of the research questions [16]. Firstly, the study draws upon interviews with 13 key respondents in the field of health and safety to inform the research design. Secondly, the study is based upon in depth case studies of 12 unionised organisations in eight key sectors represented by a total of seven trade unions (Table 1). Thirdly, an online survey of TUC Health and Safety representatives, designed by the authors, was distributed by the TUC between October and November 2021.

Deploying mixed methods generates meaningful results by identifying convergence and divergence between datasets collected from different samples [17,18]. On the one hand, the survey of trade union H&S representatives captured reported changes in health and safety structures and the extent of engagement with management over risk during COVID-19. On the other hand, the case studies offered insights into organisational policies and practice, but, above all, the agency of union H&S representatives at the micro level, something that is often elusive in the literature. The mixed methods approach allows for exploration of the dynamic between OHS infrastructures and union organisation and activity [9].

Interviews with key respondents helped to identify 12 appropriate case studies with trade union representation across sectors. These experts included representatives of the Health and Safety Executive, Institute of Occupational Safety and Health, Chartered Institute of Personnel Development, Pensions and Investment Research Consultancy, the UK Hazards Campaign, the TUC and other national union health and safety officers, and key academics in the field of occupational health, including from New Zealand. Interviews focused on the identification of risk, good practice, and thoughts on the wider UK OHS regime. 

In each case study, relevant documentation including organisational risk assessments were analysed, and in depth semi-structured interviews with three respondents were conducted (where possible), including health and safety representatives, Human Resource managers, and operational managers, with a total of 34 interviews. These interviews covered arrangements for worker representation on OHS, the impact of COVID-19 on these arrangements, the organisation of work, and the factors that facilitated or constrained representation in risk assessment and health and safety. They were largely remote via Microsoft Teams (or comparable) or, where not possible, via mobile telephone. The interviews with both OHS experts and case study respondents were conducted by three of the authors and lasted between 60 and 90 minutes. All interviews were recorded and transcribed with the consent of the participants, with reassurances on anonymity and confidentiality. The research team members independently coded a sample of transcripts using NVIVO and then jointly developed a coding template. Template analysis provided the basis for a collective thematic approach to the texts, which was iterative and encouraged the development of themes around the richest data and without an explicit distinction between descriptive and interpretive themes [19]. Key themes were the range of representative arrangements during the pandemic, the importance of representative structures as a basis for engagement on OHS, management appreciation of union health and safety expertise and resources, unilateral action by union representatives, and their importance in addressing cultures of denial, OHS as an arena for contestation, and the exclusion of groups of workers from representation. Here, template analysis was underpinned by a realist position that allowed reflection on the survey hypotheses. Analysis was validated by the presentation of initial findings to a workshop of research participants.

In 8 of the 12 case studies, workers were deemed to be essential workers and worked throughout COVID-19. In the construction sector, after the initial closure of sites, non-essential workers worked throughout. In the finance sector, employees largely worked from home. The case studies unveiled a range of hazards and risks for those working throughout COVID-19, of which social distancing was perceived as the most challenging. It is a particular issue in the case study workplaces—bus and train depots, supermarkets, and production and distribution settings. Key risks arose from initial shortages of PPE, inadequate cleaning regimes, and the (re)organisation of work in the light of social distancing, particularly if productivity or performance targets were not adapted. Sick pay emerged as a major issue during COVID-19, with limited access to occupational sick pay, and Statutory Sick Pay (SSP) an inadequate replacement. Reliance on SSP inhibited compliance with rules on self-isolation.

The survey elicited 648 usable responses (from a total of 810 responses), excluding those that were incomplete and those that were not from union H&S representatives. Respondents were from 20 TUC affiliated trade unions. In total, approaching half (43%) worked in the public sector, with 27% in transport and storage, 10% in education, 6% in wholesale and retail, and 4% in manufacturing. The sample comprises 69% male, 25% female, and 0.9% non-binary reps (with the remainder not indicating), while only 6% identified as Black and Minority Ethnic (BME). The majority of TUC H&S representatives reported that their members were deemed essential workers and worked throughout the pandemic. 

### Measures

The frequency of engagement between trade union representatives and managers is measured by two items that included increased formal and informal engagement. The response format was: 0 (decreased), 1 (the same), or 2 (increased).

The formal involvement of trade union representatives is measured by negotiation or consultation over risk assessment and PPE. Items were answered in a format of 0 (neither consulted nor negotiated), 1 (consulted), or 2 (negotiated).

Increased H&S representation is measured by an increase in the number of H&S representatives at the workplace or organisational level. Items were answered in a format of 0 (decreased), 1 (the same), or 2 (increased). 

The perception of risk is measured by agreement/disagreement with the statements, ‘The risk that the workforce faces in my workplace as a result of COVID-19 is now relatively low’, and ‘I feel confident that my employer can protect the health of the workers in future waves of COVID-19 or pandemics’ on a five-point scale from 1 (strongly disagree) to 5 (strongly agree), with a midpoint of neither agree or disagree.

The paper reports descriptive statistics and Table 2 shows the results of correlations testing the hypotheses.

## 4. Findings

Against a legacy of deregulation of OHS, one proposition is that the pandemic necessarily revitalised pluralism and the joint regulation of health and safety in organisations with existing collective structures. Another proposition is that command and control mechanisms may have overridden worker representation, while marginalisation and/or organisational lethargy could have provoked contestation and autonomous responses by union representatives. The findings firstly report the survey analysis and then move on to show how the case study material illuminates the outcome of the hypotheses.

### 4.1. The Survey

#### 4.1.1. Representation in Practice

One area of inquiry was the extent to which the pandemic impacted workplace health and safety representation. As expected from a TUC survey, the majority of respondents reported the presence of union health and safety representation before COVID-19, including union H&S representatives at the workplace level (77%) or organisational level (36%), with 11 per cent reporting non-union workplace representatives. By November 2021, those reporting union representatives at workplace level increased to 95% (an 18% increase) and at the organisational level to 45% (a 9% increase), with a 4% increase in non-union workplace representatives. Nearly one third thus reported an expansion of health and safety representation after COVID-19. In terms of actual numbers of representatives, over one in ten respondents reported an increase in the number of H&S representatives at the workplace level (9%) or the organisational level (5%). Three quarters of the respondents indicated no changes to the number of representatives at the workplace level (76%) and a majority did so at the organisational level (90%). Nearly one in five (17%) reported increased union membership. Smaller proportions, however, reported decreases in representatives and membership. 

Health and safety committees were less frequent than representatives. Overall, before COVID-19, one third (34%) of TUC H&S reps reported the existence of a health and safety committee with union representation at the organisational or the workplace level. Just under one in five (19%) stated there was a health and safety committee in place following the pandemic, taking the overall figure to 53%. 

A small proportion of respondents reported an increase in the frequency of negotiation (13%) or consultation (17%). The survey suggests union representatives played a role in a wide range of health and safety issues, including negotiation over sick pay (31%), risk assessment (29%), work arrangements (27%), PPE provision (21%), and vaccination policy (17%). Where changes in workplace policies were reported, they were considered to be a result of union pressure on a number of issues, including attendance policy (29%), sickness absence (20%), and pay (13%). Over one in ten representatives (13%) said there was more formal engagement with management over health and safety and just over one in five (21%) said that there was more informal engagement.

Just under half of TUC H&S representatives (48%) agreed that their employer was supportive of their health and safety roles—with no difference by sector and proportions similar for dedicated union safety representatives (51%) and for general union representatives (55%). At the time when the survey was circulated in November 2021, only 15% of the respondents agreed with the statement that the level of COVID-19 related health and safety risk faced by their workforce was relatively low (14% in the public and 17% in the private sector), and over two thirds (68%) disagreed. As above, around a quarter of those surveyed reported an increase in formal or informal engagement over health and safety with management, and one third (30%) of representatives reported a decrease in the frequency of negotiation or consultation over health and safety; this may suggest formal bodies meeting less frequently during the pandemic. At the same time, around one quarter reported that changes in terms and conditions were made despite union opposition, including over attendance policy (27%) and sick pay (25%). 

#### 4.1.2. The Relationship between Structures and Organisation

The hypotheses explore the relationship between structures and organisation, suggesting that structure is important but does not constrain representative activity. Table 3 shows that there is a significant relationship between the presence of a union H&S committee and increased H&S representation during COVID-19 (r = 0.338, *p* < 0.01), supporting H1. In relation to formal interaction, the presence of a H&S committee with trade union representation is positively associated with negotiation or consultation over OHS issues during the pandemic (r = 0.225, *p* < 0.01), supporting H2a. At the same time, there is no association between increased union H&S representation and negotiation or consultation over OHS issues during the pandemic, and H2b was unsupported (r = 0.123, *p* < 0.01). The presence of a H&S committee with union representatives is not associated with the frequency of (less formal) engagement between union health and safety representatives and managers over OHS issues (r = 0.127, *p* > 0.05), and H3a was not confirmed. However, increased H&S representation was found to significantly correlate with union engagement with management (r = −0.079, *p* < 0.05), confirming H3b. These findings suggest that union representative activity may be independent of structures.

H4a was also rejected, indicating that the presence of HS representative structures made no difference to H&S representatives’ perception of OHS risk during the pandemic (r = 0.115, *p* > 0.05). Similarly, H4b was unsupported, as no significant correlation was found between increased H&S representation and perceived risk (r = −0.068, *p* > 0.5). As might be expected H5a was supported, showing a positive association between union representation in HS committees and perceived managerial support for representatives (r = 0.140, *p* < 0.05). H5b was also confirmed, showing a negative association between increased H&S representation and perceived managerial support (r = −0.150, *p* < 0.05). The survey results indicate that on certain outcomes, representative activity was independent of organisational or workplace structures, particularly managerial support and negotiation, and engagement. The results do not suggest the incorporation of H&S representatives.

### 4.2. The Case Studies

The TUC survey suggests pluralist practice in OHS with joint regulation stemming from representative structures, but also the proactive role of H&S representatives. The case studies allow exploration of the role of health and safety structures, and they illustrate and confirm the agency of union representatives, including in contesting OHS. They allow reflection on the wider research questions and the tension between incorporation and unilateral action.

#### 4.2.1. The Necessity of Pluralism in a Pandemic?

In the case studies, tri-partite responses under the auspices of Transport for London (TFL) were deemed important in the underground (Tube case study) and bus sectors. For example, a bus depot representative reported that the ten-passenger limit introduced by TFL was driven by the union. Unions also ensured that all directly employed workers got full sick pay during COVID-19, including new starters who would not normally have been eligible. A regional officer commented: 

‘We made a case to TFL and the bus operators at the tripartite meeting that we were actually encouraging people to come to work. So, one of the things that we did get in place and we couldn’t get full pay for everyone, but we made sure that even new starters, where they weren’t entitled to sick pay, were paid sick pay to help them and to encourage them to isolate. And I think really our main concern there was that bus drivers were coming to work even though they thought they might have had the virus because they couldn’t afford not to. So we got TFL and the bus operators to put something in place to at least make sure that these people could eat while they were off’.

Some managers and union safety representatives reported an improvement in their working relationships during the pandemic with increased frequency of both formal and informal interactions arising from a shared sense of urgency in addressing COVID-specific issues. For one rep in finance: 

‘Since COVID, it’s been a good year, the amount of consultation and meetings that I’ve had with the health and safety people have been ramped up significantly. There was a time at the beginning practically every day, but with a scheduled meeting once a week with the Head of Health and Safety. And even now we have a meeting every other week, apart from speaking during the week if there’s an issue or something to resolve. So the communications I’d say have been very good’.

Management recognised the unique expertise H&S reps possessed, built on accredited training and union resources (‘knowledge activists’). They also acknowledged the communication role that union safety representatives played during the pandemic, particularly in the context of rapid changes in government guidance. A manager in one finance organisation reported that H&S reps facilitated staff feedback as well as communicating changes to employees:

‘I think we had a really strong and consistent management message, but I think where I see the value is, in terms of the union [they] do regularly communicate directly with their members. And their members are our employees and I think that’s the first thing that we’d always to be mindful of is that actually it can be another voice that provides that reassurance’.

Management of both case study organisations in finance were positive about union involvement in consultation over health and safety issues. In one, H&S representatives exerted pressure on the employer to revise the risk assessment on more than ten occasions in response to up-to-date information about the pandemic and spikes in infection. These representatives reported that they had made the case for avoiding layoffs and furlough given the importance of retaining some flexibility in workforce levels to manage uncertainties in new working arrangements. They also identified hazards and risks arising from the homeworking environment, and they successfully negotiated with the employer to provide employees with surge protection extension leads following a house fire arising from a member’s faulty equipment. 

#### 4.2.2. The Embeddedness of H&S Representation

Whilst existing representative infrastructures varied between case studies, they provided a necessary basis for informal and frequent dialogue with managers, as articulated by a Health and Safety rep for the underground:

‘The structure was in place already, so we already had a good structure in place and that structure was able to basically hit the ground running, to use that horrible term. We knew what to do so once we were developing the information for our members, we had the structure and the network of reps to make sure that information was getting out to everybody. So we were quite fortunate in that way’.

The case studies reflected the embeddedness of H&S representatives in the labour process and their intimate knowledge of how the organisation of work created risk, particularly in terms of social distancing and how it could be undermined by productivity targets. Reps described themselves as the ‘eyes and ears’ of the workforce in the workplace, as a rep in maternity services explained:

‘I think having local reps on the ground that see what happens operationally has been absolutely key in terms of health and safety. I think without that we wouldn’t have got a lot of the things in place for staff and patients that we do have. I think we are the eyes and ears really of the organisation and the workforce that can raise things to management’.

She also described the importance of talking to staff in the workplace to identify key issues:

‘I tend to think that most of the actions I take away from safety inspections come from what I hear rather than what I see. Because if a manager knows you’re coming to do a walk round, everything is going be put away in the right cupboard, everything is going to be looking as great as it can be. But it’s your conversations with people that tell you about what normally happens as opposed to what’s happening today perhaps. So yes, most of them came from members, but then I’ve taken an issue away on behalf of members and taken it to Health and Safety’.

In the context of the pandemic H&S rep roles extended beyond the workplace to liaising with the families of members and colleagues who contracted or died of COVID-19 and this was particularly true of sectors with high mortality rates, as in buses: 

‘Well, I would say the experience for a bus driver during this pandemic was very scary. Having seen it firsthand and know that a hearse had driven through our garage because on the day of the funeral I arranged for his body to be driven through the garage. I had all levels of management in standing, we formed a circle and they drove through. On the day of his funeral. I arranged with his son, I’d never met him before, I was given the opportunity, or the responsibility, of being the family liaison officer which I’ve never done before’.

#### 4.2.3. Representation in Practice—Incorporation?

In enforcing organisational policies on COVID-19, union representatives could be defined as being incorporated into systems controlled by management [8]. Unions regulated risk in the workplace, and this role is potentially controversial as it may be seen as policing the workforce. In the two construction case studies, positive relationships with management were reported on large flagship construction projects. Here, full-time seconded union Health and Safety Convenors, employed by the main contractors but appointed by the union, operated to represent worker interests on health and safety in a ‘partnership’ arrangement. At one construction site, the union helped to manage bus queues, with H&S representatives appointed as COVID-19 marshals and thus part of COVID-19 teams that would walk sites with security guards and challenge workers on mask wearing and social distancing. A convenor reported some tensions:

‘Because everybody is trying to clock out and everybody is trying to get on a bus, we had COVID marshals trying to control these queues and control the social distancing. People were getting a bit upset and I wouldn’t say people were being abusive, but bad language was used and things like that. And then that did lead to a couple of disagreements in the union office. But I think they were sorted out and eventually people treated the COVID marshals with respect and it was explained why they were there, what their role was. And actually they were keeping people in work and keeping them safe. And it didn’t last long, a couple of weeks and it was sorted’. 

Similarly, at the other construction site, the contractor appointed COVID-19 marshals and here they were described as being identified more with security. The convenor emphasised the tension between a policing and educational role:

‘We don’t want any altercations, we don’t hit them with a stick we just take their name and say ”well come on, this is how it’s got to be”. It’s an educational thing, it’s not something you’d do with a stick. This is not the norm and it takes people time to get used to things. So the shift patterns, then we cubicled all the changing rooms, cubicled all the canteens. So it’s not particularly, how can I put it, it’s not a great way to be, but they’re safe. Before, the canteens were a hive of activity and fun and now it’s very isolated, but you’re safe. The one way systems, we wear masks in all the canteen’s’.

In one bus company, H&S safety representatives were stood down full time to help manage social distancing in the garages and to be on hand if anybody had any concerns. In the second bus company, a convenor described how reps had a monitoring role to ensure workers were self-isolating and social distancing, but also ensured they were supportive: 

‘So, some of [the reps] were coming in during the day and nights and they were mixing it up to catch as many drivers and engineers, and anybody else, to social distance, make sure they were cleaning their hands, giving them some reassurance that there was something going on and there were people there to support them. If they needed any help or advice, or somebody to go to and talk to about anything, there was always somebody there’.

In food production union reps challenged complacency:

‘People when they’re falling sick, they don’t realise that they’ve got symptoms and they carry on mixing with people, don’t keep their social distancing. People become complacent quite often; I’ve noticed that everybody wears masks all the time and when they come up to talk to somebody they take their masks off. We say “keep the mask on, that’s the time when you need to keep them on, not when you’re just walking around”, that sort of stuff and you have to just keep reminding people “that’s the time when you need to keep your mask on”’.

It was reported that one rep had resigned because he found his ‘blood racing’ when he had to challenge the behaviour of some colleagues. In food retail the social distancing champions had customer service backgrounds and were from outside the company, again questioning the efficacy of using union reps in a policing role. 

Above all, regulating health and safety risk was dependent on the trust that union members and the wider workforce had in their workplace representatives, as one transport union officer commented:

‘Although the management might not like me saying this, they know that there is a DNA based trust with the trade union between the staff and us that they just don’t really have with the employer. It’s a different relationship. So, they know, the management know that the staff will listen to us as much as they will listen to them, and because of that they needed us. And they pretty much said, “We can’t do this without you, so help us”. And we said, “We will help you on the right terms”. And so, they were learning as they went along and they needed us to help them do that learning’.

In construction, a union convenor similarly described how the workers felt more able to communicate their views to union representatives than managers:

‘And the reps definitely took a hiding from the workforce because they feel that they can speak to their rep before the manager. It’s somebody on your level who is in the same position as you who will understand things better. And everybody was worried, so it was very busy for us. I think the reps, they probably put the best ideas forward, they really worked hard to maintain social distancing. We came in early, we stayed late, we were the people at bus stops. The safety reps did a fantastic job. Massive. If the safety reps weren’t there, I personally don’t think we would have been as successful as we have been. So, we just raised the concerns of the workforce through the channels. There’s 11 or 12 platforms, day and nightshift and there’s a rep on each platform. One of the platforms has got 700 employees on there and they’re multinational, loads of people have got vulnerable families and their own health issues. So, all of these things were coming through the reps into our safety rep meeting’. 

While the role of unions in enforcing organisational policies on COVID-19 could be interpreted as incorporation, it also gave rise to frustration and tensions where policies were not applied by managers on the ground. At a national supermarket, while there were no collective bargaining structures, there were union H&S reps who played a key role in enforcing organisational policies:

‘They were quite good because [the supermarket] have got really got some excellent policies, but the management here don’t always put it into force. We as a union have to enforce it and bring it up and remind them this is what’s going on. And then they try and say well it’s not corporate or we’re only doing what the government guidelines are and that’s including the health and safety part, let alone the sickness’.

Similarly, another union representative highlighted a disparity between organisational and managerial responses and the importance of constant union pressure from below, but also of support from senior management in overcoming managerial resistance at the local level: 

‘It’s largely because we as a union shouted very loudly and made a lot of fuss to make sure that we were involved. We are supposed to be involved in consultations anyway and I’m involved on the overall steering committee, so I was pulled into meetings. But when it came down to each specific location I made a lot of fuss about making sure that the local safety reps had been included in the discussions about what was happening on each individual site. But we have had the backing of the directors basically saying “do it, involve [the union], talk to them, include them”. So, it’s been pushed from up above, so I think we’ve done really quite well with that’. 

The concept of incorporation is fraught, since it suggests both a compromise of unions’ independent role in representing members and a willingness to endorse managerial objectives and interests at the expense of members. The evidence implies a more complex and nuanced relationship borne of union reps’ understanding of the dangers of COVID-19, with its severity re-affirming mutuality on OHS, albeit in the context of employer concern for productivity and profit. To the extent that managements might develop policies that representatives saw as protecting their members, they would endorse them, but this does not constitute incorporation. The case studies indicate that reps negotiated and mediated organisational policies, and, at times, perceived and acted upon a shared interest. At the same time, this process produced tensions and contestation, not only between management and unions, but also between unions and the workforce. Respondents reported that one of their most important roles during COVID-19 was in overcoming what was defined as ‘a culture of denial’ amongst the workforce. 

#### 4.2.4. Tensions and Contestation

In health trusts, there was criticism of the deployment of ‘command and control’ responses to the pandemic that could marginalise trade unions and discourage workers from raising concerns and questioning the health and safety practices of senior management. In one ambulance trust, ‘command and control’ was seen as potentially imbricating ‘learned behaviours’ in response to emergencies, embedded in hierarchical structures that engendered a lack of trust of those in both leadership and managerial roles. A similar command and control strategy was reported in the case study of maternity services. Initially, the Trust invoked an emergency response described as ‘Operational Command and Control’, with ‘non-essential meetings’, including the joint negotiating council, stood down, and with health and safety incorporated in the Command-and-Control meeting. The number of accredited health and safety reps increased substantially during the period of the pandemic, and one expressed dissatisfaction at the consequent removal of opportunities to represent members:

‘Some of the meetings where we got the opportunity to have a voice on behalf of members were stood down. And that was quite challenging for us because we felt as unions it was a time when actually our involvement was even more vital. So, for example, for a couple of months the JNC seat—so the Joint Negotiation Council was stood down and that’s where we have an opportunity to hold managers to account on decisions that are being made that affect staff. So, there were a lot of changes obviously coming at a rapid pace because of the evolving Coronavirus situation, but yet we had less opportunity to offer any constructive challenge as to how that was affecting staff’.

Following a formal grievance raised by the unions, joint committees, including health and safety, were reinstated. Bi-monthly meetings of the Health and Safety Committee took place online, and following these meetings, the health and safety rep met with an Associate Director daily to brief her on specific issues. 

Across the case studies, contractual variation as a result of privatisation and sub-contracting, and particularly the increased deployment of agency workers in the pandemic, undermined OHS representation. Workers on non-standard contracts had limited access to workplace representation and employment rights: their contractual status limited entitlement to occupational sick pay, but also SSP. Participants highlighted specific health and safety issues faced by agency workers, often migrants, particularly in health, transport and food production, where they worked on production lines and as cleaners and porters or security staff. At one of the case study organisations in construction, a representative reported that self-employed workers were reluctant to use the COVID-19 track and trace system as they could not afford to take sick leave.

In a distribution case study, management refused approaches by the union to represent agency staff because the agency was ‘a third party.’ However, a transport union rep reported that the union had negotiated full sick pay from day 1 for 2500 cleaners working for contractors on the underground if they were off with COVID-19 symptoms; they previously only had access to SSP:

‘So, of course if a cleaner develops symptoms and they’re on the breadline and they can’t afford to take the time off, they might have been inclined to come into work. So, we actually negotiated that TFL would pay these people their full salary if they were off with COVID. We were saying there’s no point in us taking all these measures to protect ourselves and then there’s a weaker link where people aren’t going to get paid any wages and they might be forced financially to come to work. It might be the difference between losing their accommodation or not eating that week … I really feel for the cleaners’. 

As for directly employed workers, sickness absence triggers were removed during COVID, although the company had attempted to reinstate them.

Unions were keen to ensure that COVID-19 did not become a disciplinary issue. Some organisations had disciplinary procedures for breaching health and safety measures. At one finance organisation there was a three-tier system with regard to social distancing and mask wearing. Initially, workers were made aware of the breach; on the second occasion, they were required to retake induction; and on the third, a breach was declared and they were forced to leave the building and work from home, with potential disciplinary action. A substantial breach could go straight to disciplinary action. At the other finance organisation, the rep reflected the tension between employer disciplinary measures and the role of reps in enforcing health and safety amongst colleagues:

‘Obviously we don’t want to go in there and get people into trouble, but we also need to make sure that if people are going to work then they need to be as safe as possible. And if people are not wearing masks, we pick it up, not by name-we wouldn’t name them-but we probably would ask them why they weren’t wearing a mask, which is always a bit tricky. And I know a colleague in Cardiff has actually confronted a couple of people about the fact they weren’t wearing a mask’.

At one of the large flagship construction sites, a regional officer reported that workers refusing to wear face masks or to be randomly tested could be removed from site and a rep recounted that repeat offenders were dismissed. Elsewhere in construction, a regional officer stated that one employer had intended to discipline workers after three instances of not wearing masks when walking to the toilet:

‘And we got that stopped by saying “well no other company in the country is doing that, people need to be wearing their mask when they’re travelling from their desk to the toilet, but equally, three strikes and you’re out seems to be extremely harsh”’.

One union rep objected to employers making health and safety an individual problem: 

‘I think the social distance element of it, the reaction from the managers was that “it’s staff’s fault”. It was almost apportioning blame if it was going wrong, it was the staff’s fault at that time, which was disappointing. Because there were some minor adjustments that they could have made to stop people from doing it. The environment is ultimately their responsibility, the people’s health and wellbeing is ultimately their responsibility. It’s not up to staff really to take ownership of that responsibility’.

#### 4.2.5. Representation in Practice—Autonomous Collective Activity?

The case studies identify a range of proactive and even autonomous collective activity by H&S representatives, including unilateral action. In construction, they advocated the use of masks before the precautionary measure was included in government guidance. In the case study of the underground, reps suggested that the employer’s response to COVID-19 was informed by experience of the Norovirus that occurred 6 years previously, during which the dangers of working in confined spaces, such as driver’s cabs, were recognised. The union had negotiated for latex gloves, bacterial wipes, and hand sanitisers during the Norovirus outbreak, and renewed these demands during COVID-19. To ensure social distancing the union negotiated occupancy levels for shared spaces, including the mess rooms, with training rooms used for overspill:

‘So, in a mess room you could have maybe up to 10, 12, 20 people all sitting having lunch at the same time. So, we removed chairs, we made sure that nobody was facing each other. We increased the ventilation in the room by opening the windows and we had notices up everywhere around the depot informing everyone of every room they went into what the occupancy level of that room was. So, to keep the numbers of people in one room down to the bare minimum we introduced another couple of mess rooms and put mess room facilities in there such as kettles and microwaves and stuff. So that was basically the initial get go, so it was occupancy levels, hand sanitiser, face masks, bacterial wipes’. 

The union also instructed members to refuse to engage in cab training that involved two people in close proximity. Following initial resistance management cancelled cab training, but also conceded on the need for improved cleaning regimes. The union ensured that drivers could book-in to work by phone, and that if they were informed at work that anyone in their household had developed symptoms, they would be relieved immediately and could go straight home without going back to the depot. Measures were then taken to ensure the drivers’ cab would not be allowed back into service until it had been deep cleaned. The union also demanded that the organisation introduce station controllers and that local building sites stagger starting times to avoid overcrowding at stations. 

In one health trust, midwives were required to come to work in their own clothes and change prior to and at the end of their shift. Changing facilities were not conducive to social distancing, resulting in staff having to come into work earlier and leave work later to stagger their changing time. The joint unions were attempting to negotiate paid changing time at the time of the interviews. They had also negotiated an additional two 15-min breaks, one in the morning and one in the afternoon for hydration, recognising the stress of wearing PPE. Union reps were involved in discussing and seeking solutions in individual cases where midwives were unable to attend work because of their own caring roles. 

In some cases, union H&S reps took unilateral action to mitigate perceived risk. For example, a convenor representing engineers on the buses marked out social distancing using black and yellow tape. He recalled regular visits to two or three garages every night before the first lockdown to explain and promote preventative measures, including the disinfection and ‘fogging’ (fumigation) of buses. He felt that the union had played an effective role during the pandemic, and highlighted the importance of union recognition:

‘I’ve got to say I’m so glad the union was there and there was a union role to be played because if it wasn’t for them a lot of things wouldn’t have happened. Forcing the masks issue, closing of the doors, then reopening of the doors. If the union weren’t there and around, none of that would have happened, none of that would have happened. So I think the union has had a major, major role in this, in a good sense and I don’t feel that the management would disagree when they said that at least there was somebody there to help us as well to put things into process. Because if we were non-unionised, people would be just running around, doing whatever they wanted. But where there was a union person there in most garages from the operating side, or the engineering side, it controlled a lot of situations. So the union has had a massive, massive role in this and they’ve helped out our company—we could have had a lot more deaths. We could have had a lot more seriousness’. 

An ambulance service rep also reported her unilateral action over social distancing:

‘For example, in the communal areas, the tables were all close together. It simply needed 1, 2, 3, 4 tables removing and the chairs removing so that we could socially distance. When I approached the manager, she said “I’ll see what estates say and they’ll probably come next week”. And I said, “no it needs to be done, well I’ll go down and do it myself”. And she just said, “oh if you wouldn’t mind”. It’s not my job, but I wasn’t prepared to wait a week for that to happen. So it was that kind of thing, it’s almost like the managers were waiting for permission to be able to change the risks that were there at that time. As a health and safety person I can remove that risk immediately if I think that there’s a risk to people’s health and wellbeing. So that’s exactly what I did’. 

Having ensured social distancing, she also described how she intervened to ensure enhanced around-the-clock cleaning routines, responding to night staff complaints that they did not see cleaners.

## 5. Discussion

Hall and Tuckers’ study of union determination to shape COVID-19 safety policy pointed to a paradox that the absence of power undermines the exercise of legal rights, yet without stronger legal rights workers are constrained in the exercise of power [9]. Analysis of the TUC survey found that, as might be expected, the presence of a Health and Safety Committee is related to perceived management support for OHS, to the frequency of negotiation with management and also to expansion in representation since COVID-19. At the same time, there is a suggestion that H&S representatives may substitute for regulatory structures—thus the presence of a Health and Safety Committee is not related to informal engagement with management nor with perceived risk in the workplace. 

The research presented here focuses on H&S reps in unionised workplaces and suggests an increase in union H&S representation following COVID-19. It corroborates the TUC’s 2020 survey of its H&S representatives, which confirmed increased recruitment of new safety reps, with 18% of those responding having been a rep for less than 1 year [20]. H&S reps were spending more time fulfilling H&S representative functions as a result of the pandemic, although only half were paid for doing so. Our more recent survey indicates some increase in health and safety committees and some increase in negotiation, but only just over half of H&S representatives reported health and safety committees in their workplace or organisation after the start of the pandemic, suggesting a structural and regulatory deficit. 

While tripartite structures in transport appeared effective, in the case of the health service command and control responses to the pandemic could override worker participation and in one case the union challenged such marginalisation. Sheratt and Dainty suggest, with regard to the construction industry, that COVID-19 put worker OHS front and centre with the potential for rethinking ‘sites as places of illness and infection, and to mitigate accordingly’—and the opportunity for a paradigm shift [21] (p. 6). In the construction case studies presented here, full-time seconded union Health and Safety Convenors, employed by the main contractors, underpinned a ‘partnership’ arrangement. However, across the sector, extensive subcontracting and ‘self-employment’ removes workers from representation. In many of the case study sectors, the increased use of agency workers (often migrant workers) meant the denial of employment rights, including access to adequate sick pay. The use of agency workers, who by definition move between workplaces potentially spreading infection, was raised by respondents as a major issue and as counter intuitive. 

The case studies demonstrate the importance of representative mechanisms as the basis for more informal day-to-day engagement over H&S during COVID-19, with the agency of workplace reps crucial. However, it could also be argued that their absence necessitates autonomous collective representation. The capture of representation is challenged by the survey where, following their experiences of COVID-19, only one in five of the representatives (19%) reported confidence in their employers to protect the health and safety of workers in future waves of pandemics, and half felt they did not have management support. As in the international literature on union responses in the pandemic [13,14,15], the case studies confirm the importance of effective and autonomous representation in reducing risks at work [11], highlighting the proactive role played by union H&S reps based on their embeddedness in the labour process and their intimate knowledge of how the organisation of work creates risk. In some cases, they were in the workplace, while senior managers were working from home. Workplace representatives exerted bottom-up pressure on management, holding management to account, enforcing organisational policies, and ensuring the immediacy of employers’ responses to the pandemic. They pressed for risk assessments, but also regulated the implementation of measures at the workplace. Health and safety representatives took unilateral action in some cases, with tension over health and safety emerging in other cases and unions taking grievances to control risk. In monitoring risk amongst the workforce, ensuring compliance with regulations and overcoming resistance to them, H&S representatives challenged what one rep called ‘a culture of denial’. Unions were keen to ensure that COVID-19 did not become a disciplinary issue, although the tension between the educational and policing role was evident. Union representatives stressed the role of education and communication based on the trust union members had in their representatives in ensuring compliance, and criticised attempts to make health and safety an individual rather than organisational or societal issue. As in the wider literature, they reported the strain placed on workplace representatives and consequent exhaustion. 

Key respondents emphasised an inadequate national infrastructure to deal with the pandemic, including confused government advice, under-reporting of workplace infection, and the weakness of the Health and Safety executive following funding cuts. Overwhelmingly, participants felt that they wanted to see the powers and capacity of the Health and Safety Executive in relation to UK workplaces both restored and strengthened, including ensuring that the Reporting of Injuries, Diseases, and Dangerous Occurrences Regulations 2013 (RIDDOR) were appropriately used and enforced. Equally, they identified the need to reform sick pay if current and future pandemics are to be mitigated and avoided. The undermining of statutory and occupational sick pay, particularly for low-paid, agency, and self-employed workers, means that workers cannot afford to shield or isolate in the event of pandemics, intensifying the risk of transmission. In the US, Ghilarducci and Farmand evidenced the detrimental impact of the absence of sick pay for frontline older workers in COVID-19 and the need for government legislation [22].

## 6. Conclusions

The pandemic necessarily revitalised pluralist responses where joint regulation existed, yet at the same time, command and control mechanisms saw attempts to marginalise worker representation in the health service. While H&S representatives mediated organisational strategies to combat COVID-19, evidence of their independent workplace activity challenges the notion of ‘captured representation’ as an absolute, lending itself to a more nuanced picture with inherent tensions. Far from being unitarist in practice, the evidence here suggests that OHS in the context of a pandemic has been contested and is an arena for industrial conflict reflecting the wider political economy. The latter embraces a legacy of deregulation and re-commodification of labour, which entails the removal of employment protections, including limitation on access to occupational sick pay and the inadequacy of SSP. In a hostile industrial relations climate and in the context of a pandemic, this research suggests the necessity, but not desirability, of autonomous collective representation of workers’ interests in health and safety [8] (p. 87). Joint regulation on the basis of independent union representation and organisation remains fundamental to the integral relationship between workplace health and safety and public health.

## Figures and Tables

**Table 1 ijerph-20-05551-t001:** Case Studies.

Case Study	Sector	Interviews Trade Union Representatives (of Which Are Workplace Representatives)	Employer Representative
Busco1 Busco2	Buses	4 (2)	1
Tube	Underground	3 (1)	
Ambulance	Health	3 (1)	2
Maternity1 Maternity2	Health	2 (1)	1
Insurance1 Insturance2	Finance	3 (2)	3
Construction1 Construction2	Construction	4 (2)	2
Supermarket	Food Retail & Distribution	2 (1)	1
Food	Food Production	2 (2)	1
Total		23	11

**Table 2 ijerph-20-05551-t002:** Results of Pearson correlations.

	Frequency of Engagement	Union Negotiation or Consultation	Perceived OHS Risk	Perceived Managerial Support	Expanded union Representation
Union HS committee	0.127	0.225 **	0.115	0.140 *	0.388 **
Expanded union representation	0.079 *	0.123 **	−0.068	−0.150 *	

* Correlation is significant at the 0.05 level (2-tailed). ** Correlation is significant at the 0.01 level (2-tailed).

**Table 3 ijerph-20-05551-t003:** The hypotheses and results.

Hypothesis	Results
**H1.** *The presence of a H&S committee with trade union representation is positively associated with increased H&S representation*.	Supported (r = 0.338, *p* < 0.01)
**H2a.** *The presence of a H&S committee with trade union representation is positively associated with negotiation or consultation over OHS issues during the pandemic*.	Supported (r = 0.225, *p* < 0.01)
**H2b.** *Increased H&S representation is negatively associated with negotiation or consultation over OHS issues during the pandemic*.	Unsupported (r = 0.123, *p* < 0.01)
**H3a.** *The presence of a H&S committee with trade union representation is positively associated with the frequency of engagement between trade union representatives and managers during the pandemic*.	Unsupported (r = 0.127, *p* > 0.05)
**H3b.** *Increased H&S representation is positively associated with the frequency of engagement between trade union representatives and managers during the pandemic*.	Supported (r = 0.079, *p* < 0.05)
**H4a.** *The presence of a health and safety committee with trade union representation is negatively associated with perceived OHS risk*.	Unsupported (r = 0.115, *p* > 0.05)
**H4b.** *Increased H&S representation is negatively associated with perceived OHS risk*.	Unsupported (r = −0.068, *p* > 0.5)
**H5a.** *The presence of a H&S committee with trade union representation is positively associated with perceived managerial support for H&S representation*.	Supported (r = 0.140, *p* < 0.05)
**H5b.** *Increased H&S representation is negatively associated with perceived managerial support for H&S representation.*	Supported (r = −0.150, *p* < 0.05)

## Data Availability

The data used in this study are managed by the authors. To access these data, please contact the authors.

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
