# Peer review of "Health and Safety Reps in COVID-19—Representation Unleashed?"

_ijerph, 2023, doi:10.3390/ijerph20085551_

Round 1
Reviewer 1 Report
some notes to improve the quality of the journal include methods and results and conclusions.
Added real contributions like workers

Reviewer 2 Report
Thank you for the opportunity to read this manuscript. The article “Health and Safety Reps in COVID-19 – representation unleashed?” aims to explore the role of UK union health and safety reps and changes to representative structures and mechanisms governing workplace and organisational Occupational Health and Safety (OHS) during COVID-19. The research question is relevant, the methodology and data (survey of TUC H&S reps and case-studies of 12 organisations in eight key sectors) are appropriate.
I would just add a couple of suggestions to improve the manuscript.
1. In the results section highlight similarities and differences between sectors/ industries.
2. In the discussion, include a comparison with results from similar studies.
Finally. please, check that the journal allows the use of ‘reps’ for representatives.
Reviewer 3 Report
Thank you for submitting your article IJERPH. The subject of your article is very interesting and timely as we reflect on learning from the peak of the pandemic. Your study has merit however as a scholarly piece of writing there are several areas which require attention before consideration for publication. I hope that the suggestions I have made below will assist you to revise your manuscript for publication.
The introduction and context do a fairly good job of setting the scene for your study although the use of acronyms writing them in full and shortening words (i.e., reps) needs attention to ensure that the reader understands what you are referring to, particularly those not located within the United Kingdom. You argue your thesis well although a deeper discussion regarding the unitarist perspective would make this theoretically stronger.
I found that the methods section was unclear and did not include enough information to fully grasp what data was collected and how. It is unclear why you chose to do interviews, case studies, and have a quantitative survey. What was your rationale for this? What was the purpose of each type of data collection methods to answering your research question? Thus, each of the three data collection methods needs to be very clearly set out and explained. Sub-headings might be helpful here.
Interview - The participants in the 13 interviews that you allude to in footnote one need to be explained in more detail in a Table or Appendices if wordcount is an issue. You also need to include the length of the interviews (average length would be acceptable) and give the reader a clear understanding of the exact purpose of the interview and at a minimum some idea of the questions asked. Were the interviews transcribed/recorded? Were they conducted by all or some of the authors? Who did the analysis of the interviews? What was the analytical approach? You also mention that one of these interviewees was from New Zealand. How was interviewing this person relevant to your research question?
Case studies - In Table 1 in the case study interviews column there are numbers in brackets – what do these represent. The acronym TUC is not explained to the reader. How did you construct the case studies? What additional data did you use to develop the case study? How were these checked for accuracy i.e., by someone from the organisation?
Survey – there is very little information about the survey. What measures were used and why? Was the study designed by the authors or was it panel data sourced by the authors.
There is no analysis section providing detail about how you analysed the survey or the approach you used to analyse the case studies or interviews. The lack of a clear analytical approach flows into the results section which struggles to integrate the qualitative and quantitative data you have collected. I was also surprised to see you have only use minimal descriptive statistics to analyse your survey. Assuming you designed this survey and collected the data it would seem to be a waste of your data. Consequently, reading the results is rather disjointed. There needs to be a stronger connection between the data. It appears that the qualitative survey is being used to confirm the findings in the qualitative analysis – is this the best way to use the data; was that your intention ?
The discussion could be strengthened with greater synthesis of the results to answer your research question. Additionally, stronger integration of the literature is needed.
Overall, the paper needs to be thoroughly edited. There are numerous grammatical errors and typos which need attention. For example, Pg 2 line 58 ‘can underplay that the workplace’; line 74 ‘capacity to exercise of their right and…’; Pg 5 lines 171 & 178 inconsistent capitalisation of COVID-19. Additionally, in Section 4. You have left the journal guidance blurb in the section. This needs to be replaced with an introduction to the results. There are also several quite poorly constructed sentences which would be addressed through editing. The sentence on page 6, line 241 seems out of place. Finally, there is some anthropomorphism in the paper that need attention.
Reviewer 4 Report
Dear Authors
I have read your paper with great interest. The paper's aims are clear and the topic is worthy of inquiry, and timely in its importance. A few points are noted below. However, I must indicate that regrettably I had to stop reading half way through the results section. The latter is due to the fact that tables are missing (Except table 1), and the results sections is too cluttered and too long to be able to follow the results, which derives from which instruments, and so on. The more minor comments I have are the following:
1. The abstract can be improved. It is confusing as is, and not much of an abstract-like content. Here, also make sure you explain abbreviations before introducing these in, importantly, the abstract. I am aware of the UK employment aspects, unionisation and TUC, but the audience may not as you can understand.
2. Under material and methods, you provide the approach to the qualitative data analysis, but not for the survey. please add this information.
3. Under material and methods also give some insight on the interview themes discussed.
4. Attempt to balance the information and length of sections, and focus on conveying key messages.
5. Please explain, why the information on pg. 3, lines 100-101 is included; why is this important
6. Add some clear visitation to manage the results section.
7. I also wonder how %s were rounded up, as it appears?
8. On pg. 4, amend the introduction under section 4, with your own input.
Round 2
Reviewer 3 Report
Thank you for your efforts to improve your manuscript. Despite some issues with the presentation and execution of your paper I believe it is an interesting and important group of studies and worthy of being published.
You have gone some way to make the paper more scientifically sound. However, there are several areas of the paper that require further detailed attention to ensure that the necessary depth of explanation is provided to convince the reader that the studies have been undertaken with the required degree of academic rigour. I have made several specific suggestions directly onto a PDF of your paper (attached). My hope was that this would be of practical use to you in improving your paper.
Reading the revised paper the main issues are now structural which I believe if addressed effectively will significantly improve the readability of the paper. The inclusion of hypotheses add to the impact of your paper however, the hypotheses should be introduced in the literature review section of the paper and be numbered consistently throughout the paper using the typical conventions. Likewise, the explanation of the survey items is necessary but should be part of the methods section.
I encourage you to read your results section and consider where you have been overly descriptive and risk losing the reader. Are there opportunities to use Tables accompanied by brief explanations and interpretations?
You mention that a template was used to analyse the interviews but you do not mention what the themes were agreed and the results do not seem to be discussed within any themes (or perhaps this is unclear).
It is inherently challenging to integrate three studies of mixed methods into one paper. You began well by indicating the initial qualitative study was a means to further develop the case studies and inform the items in the survey. This explanation could be made clearer, especially in the reporting of the results. The movement between qualitative and quantitative results becomes challenging to follow.
I wish you well in further improving your manuscript

Author Response
We have responded to all reviewers' comments and proposed changes, we have added a table of hypotheses and also separated the quantitative and qualitative findings - we hope that this makes the findings clearer.